# Development of a Strategy for Enhancing the Biomass Growth and Lipid Accumulation of *Chlorella* sp. UJ-3 Using Magnetic Fe_3_O_4_ Nanoparticles

**DOI:** 10.3390/nano11112802

**Published:** 2021-10-22

**Authors:** Feng Wang, Tingting Liu, Wen Guan, Ling Xu, Shuhao Huo, Anzou Ma, Guoqiang Zhuang, Norman Terry

**Affiliations:** 1School of Food and Biological Engineering, Jiangsu University, Zhenjiang 212013, China; 2221918053@stmail.ujs.edu.cn (T.L.); ghwconan@163.com (W.G.); lxu@ujs.edu.cn (L.X.); huo@ujs.edu.cn (S.H.); 2Key Laboratory of Environmental Biotechnology, Research Center for Eco-Environmental Sciences, Chinese Academy of Sciences, Beijing 100085, China; gqzhuang@rcees.ac.cn; 3Department of Plant and Microbial Biology, University of California, Berkeley, CA 94720, USA; nterry@berkeley.edu

**Keywords:** Fe_3_O_4_ nanoparticles, chlorella, biomass growth, lipid production, fatty acid

## Abstract

In this study, magnetic Fe_3_O_4_ nanoparticles (NPs) were used as an effective enhancer to increase the biomass and total lipid production of *Chlorella* sp. UJ-3. It was found that the biomass of algal cells increased significantly when they were exposed to low concentrations of Fe_3_O_4_ NPs (20 mg/L), while the best total lipid content of algal cells was achieved when they were exposed to high concentrations of Fe_3_O_4_ NPs (100 mg/L). Therefore, we established a strategy to promote the growth and lipid accumulation of microalgae by initially exposing the algal cells to low concentrations of Fe_3_O_4_ NPs and then treating them with an increased concentration of Fe_3_O_4_ NPs after 12 days of culture. For this strategy, the biomass and total lipid production of algal cells increased by 50% and 108.7%, respectively, compared to the untreated control. The increase in lipid production and change in the fatty acid composition of Chlorella cells were found to help them to cope with the increased number of reactive oxygen species produced due to oxidative stress in alga cells after the addition of Fe_3_O_4_ NPs. This study provided a highly efficient way to improve the lipid production of microalgae using nanoparticles.

## 1. Introduction

With the rapid development of nanotechnology in recent years, nanoparticles (NPs) have exhibited various special properties due to their extremely small size, such as the surface effect, quantum size effect, small size effect, and macroscopic quantum tunneling effect. NPs have been widely used in the fields of optics, electronic biomedicine, food, cosmetics, environmental energy, and chemistry [1,2,3]. Additionally, the biological effects of NPs have also attracted the interest of researchers. It was found that the use of an appropriate concentration of NPs was beneficial to the growth of biological cells and the synthesis of target products [4,5]. Recently, it was reported that nanoparticles could promote the growth of barley, wheat, and *Zea mays* [6,7,8]. TiO_2_ NPs were found to have a positive effect on the production of unsaturated fatty acids (UFAs) in *Pichia pastoris* to fight against oxidative stress [9]. Trace amounts of Se NPs were reported to promote the growth of *Aspergillus niger* and *Candida albicans* in a dose-dependent manner and had a strong stimulatory effect on the growth of *Aspergillus niger* [10]. It was found that carbon nanotubes, nano-Fe_2_O_3_, and nano-MgO could increase the content of chlorophyll, protein, and lipid in *Scenedesmus obliquus* [5]. In addition, SiO_2_ NPs were able to enhance the growth and lipid production of *Chlorella vulgaris* [11], while SiC and g-C_3_N_4_ NPs improved the biomass and lipid accumulation of *Scenedesmus* sp. [12]. However, in the study on the effect of Fe_3_O_4_ NPs on *Picochlorum* sp., Hazeem et al. observed a decrease in cell concentration during the exponential growth phase of algal cells after the addition of NPs, while the growth of algal cells was promoted in the stabilization and decay phase [13]. Zero-valent iron NPs were applied in the culture of the eustigmatophycean algae and cyanobacteria. The results showed that trace concentrations of zero-valent iron NPs significantly increased lipid accumulation in all microalgae [4]. Evidently, the biological effects of NPs possessed broad application prospects for promoting the biomass growth and synthesis of metabolites. The related research is still in progress, and the relevant mechanism has remained unclear until now. Further studies are expected to be carried out and an effective strategy will be developed and established.

Microalgae are widely distributed across the land and in the ocean. They possess a high photosynthetic efficiency and are rich in nutrients. They have good application prospects in the pharmaceutical industry, food industry, and for animal feed and bioenergy [12,14,15]. *Chlorella* sp., as a nutrient-rich microalgae, is considered to be one of the most promising commercial algae candidates. *Chlorella* sp. have also been considered as one of the candidates for biodiesel production due to their high lipid content [16]. The high cost of microalgae cultivation is one of the key factors inhibiting the industrial application of microalgae. Increasing biomass production and/or target product content can be an alternative means to achieve low-cost microalgae production. Lipid accumulation and fatty acid composition in microalgal cells changed when microalgae were exposed to physical or chemical stimuli [17]. Recently, researchers have explored different means to promote the growth of *Chlorella* sp. and enhance their lipid productivity. Fan et al. investigated the growth and lipid accumulation of *Chlorella pyrenoidosa* under different concentrations of CO_2_, and it was found that better algal biomass growth and a high total lipid content were achieved at 5% CO_2_ [16]. Microalgae exhibited a higher photosynthetic efficiency and accumulated more neutral lipids when supplied with high-dose CO_2_ [18]. The growth and total oil production of *Chlorella vulgaris* could be significantly increased by treatment with indole-3-acetic acid and jasmonic acid during the early growth phase of microalgal cells [19,20]. Plant hormones such as gibberellin, cytokinin, and abscisic acid possessed similar effects [21]. Under nitrogen starvation conditions, lipid accumulation was induced in the cells of *Chlorella* sp. [22,23]. In addition, a mutant strain of *C. vulgaris* obtained by random mutagenesis using ethyl methanesulfonate exhibited a high biomass and high lipid content [24].

Fe_3_O_4_ nanoparticles (Fe_3_O_4_ NPs) are less toxic to organisms, and their dissociated iron ions are essential micronutrients for phytoplankton because they can participate in cell photosynthesis and respiration [25,26,27,28]. The objective of the study was to test the regulation of Fe_3_O_4_ NPs on the growth and the lipid accumulation of *Chlorella* sp. UJ-3. Additionally, a new strategy eventually needs to be established to enhance the growth and lipid accumulation of *Chlorella* sp. UJ-3 under controlled concentrations of Fe_3_O_4_ NPs. Furthermore, the total lipid production, fatty acid composition, oxidative stress, and antioxidant enzyme activities in algal cells were investigated in this study.

## 2. Materials and Methods

### 2.1. Microalgae Strain and Culture Conditions

*Chlorella* sp. UJ-3 was obtained from the School of Food and Biological Engineering, Jiangsu University (Zhenjiang, China). The strains were grown in 250 mL Erlenmeyer flasks containing 100 mL of BG-11 medium, which was composed of 1.5 g/L NaNO_3_, 0.04 g/L K_2_HPO_4_·3H_2_O, 0.075 g/L MgSO_4_·7H_2_O, 0.02 g/L Na_2_CO_3_, 0.036 g/L CaCl_2_·2H_2_O, 0.001 g/L Na_2_EDTA, 0.006 g/L citric acid, 0.006 g/L ammonium ferric citrate, and 1 mL microelements solution (2.86 g/L H_3_BO_3_, 1.81 g/L MnCl_2_·4H_2_O, 0.22 g/L ZnSO_4_·7H_2_O, 0.39 g/L Na_2_MoO_4_·2H_2_O, 0.08 g/L CuSO_4_·5H_2_O and 0.05 g/L Co(NO_3_)_2_·6H_2_O) [29]. All chemicals of analytical grade were obtained from Sinopharm Chemical Reagent Co., Ltd. (Shanghai, China). All cultures were incubated at 25 °C with continuous shaking at 150 rpm under a light intensity of 100 μmol/(m^2^·s) (16 h light/8 h dark cycle) in an incubator equipped with fluorescent lamps (HYL-C2, Taicang Qiangle Experimental Equipement Co., Ltd., Taicang, China).

### 2.2. Synthesis of Nanoparticles

Totals of 2.703g FeCl_3_·6H_2_O and 0.994g FeCl_2_·4H_2_O were dissolved in 100 mL of deionized water at 80 °C, then 10 mL of ammonium hydroxide was added and the mixture was stirred at high speed for 30 min. The precipitate was washed with sterile water three times, then underwent ultrasonic treatment (100 W, 40 kHz) for 20 min before each use to prepare a magnetic Fe_3_O_4_ nanoparticle suspension. The particle diameter obtained from the transmission electron microscopy (TEM, HITACHI H-7650, Hitachi of Japan Ltd., Tokyo, Japan) image was less than 10 nm (Figure 1).

### 2.3. Experimental Design

The microalgae cells were inoculated into fresh BG-11 medium with an initial biomass of around 0.2 g/L dry cell weight. Suspensions of Fe_3_O_4_ NPs with different concentrations (10, 30, 50, 80, 100, and 200 mg/L) were added into the medium before inoculating the algae cells. The cultivation conditions were as described above (Section 2.1). Three biological replicates of each condition were carried out. After 21 days of cultivation, the biomass and total lipid content of each treatment of algal cultures were determined. Based on a wide range of screening results, it was found that Fe_3_O_4_ NP concentrations had different effects on the growth and lipid accumulation of *Chlorella* sp. UJ-3. Then, to obtain the concentration that best promoted algae biomass growth, the dosage of NP was optimized in a low concentration range (0–40 mg/L) and the NP dosage in high concentration range (60–160 mg/L) was investigated in order to obtain the best NP concentration for lipid accumulation. Subsequently, a new optimal combination strategy was obtained to increase the total lipid production by adding NP of low concentration at the initial stage and increasing NP concentration to a high level after the maximal specific growth rate was obtained.

### 2.4. Determination of Biomass and Total Lipids Content

The microalgae cells were obtained by centrifugation at 10,000 rpm for 10 min, dried by a lyophilizer (FreeZone^®^, Thermo Fisher Scientific, Waltham, MA, USA), and measured gravimetrically. The algal biomass was the weight of the dried cells subtracted from the weight of the nanoparticles added into the medium. Intracellular total lipids were extracted from a certain amount of algal cells with a mixture of methanol/chloroform (2:1, *v*/*v*) [30]. The chloroform layer containing total lipids was removed through nitrogen and the amount of total extracted lipid was determined gravimetrically. The lipid content was calculated and expressed as its quality percentage in microalgae biomass. Finally, the lipid production can be calculated based on the biomass production and lipid content. 

The calculation of specific growth rate and was carried out in the software of Origin 2019 (Origin 2019, OriginLab, Waltham, MA, USA). For this purpose, the biomass curve along with the culture time was fitted by Logistic model and then the 1st derivative of Logistic fit was applied. After that, the change curve of specific growth rate with respect to the culture time was achieved and the maximal specific growth rate can be obtained from the curve. 

Lipid productivity was calculated using the following equation:Lipid productivity (P, mg/L/day) = (χ_2_ − χ_1_)/(t_2_ − t_1_)(1)
where χ_1_ and χ_2_ were the lipid production (mg/L) on days t_1_ and t_2_, respectively.

The lipid productivity was calculated for each 3-day interval and the maximal lipid productivity will be obtained after all calculations.

### 2.5. Fatty Acids Analysis

The fatty acid content of the microalgae was estimated as the amount of fatty acid methyl esters (FAMEs), which was obtained using the direct esterification method. The obtained crude lipid was saponified with KOH-CH_3_OH and then FAMEs were prepared by methylene esterification with boron trifluoride in methanol and extracted with n-hexane [31].

The qualitative and quantitative analyses of the lipid sample were carried out using a GC-2030AF gas chromatograph (Shimadzu, Tokyo, Japan) equipped with a DB-Wax capillary column (30 cm, 0.32 mm ID, 0.5 μm film thickness) and a flame ionization detector (FID). The temperature program was as follows: initially kept 120 °C for 3 min; then increased to 190 °C at 10 °C/min, where the temperature was maintained for 1 min; then increased to 230 °C at 2 °C/min and held there for 15 min with a split ratio of 10:1. Nitrogen was used as a carrier gas at a flow rate of 3 mL/min. Fatty acids were identified by comparing the peak and retention time of the reference standard 21-component fatty acid methyl ester mix (NU-CHEK-PREP, INC). The relative and absolute contents of fatty acids were quantified by the area normalization method using methyl heptadecanoate purchased from Sigma-Aldrich (Merck & Co., Kenilworth, NJ, USA) as an internal standard.

### 2.6. Oxidative Stress Assessment and Determination of Antioxidant Enzyme Activities

The algal cultures were collected by centrifugation at 10,000 rpm for 10 min at 4 °C, then the pellet was resuspended in 2 mL of 0.05 M sodium phosphate buffer solution (PBS, pH 7.8) and immediately disrupted by sonication (JY92-IIDN, Ningbo Scientz Biotechnology CO., LTD, Ningbo, China) for 5 min with a 3 s pause after each 3 s pulse in an ice bath. The cell homogenate was centrifuged at 4000 rpm for 10 min at 4 °C and the supernatant was recovered for the assessment of enzyme activities, the malondialdehyde (MDA) level, and the hydrogen peroxide (H_2_O_2_) content. The soluble protein content was determined using the method of Bradford [32]. The H_2_O_2_ content was measured by a commercial assay kit (Nanjing Jiancheng Biology Engineering Institute, Nanjing, China) following the manufacturer’s instructions and the result was expressed as mmol H_2_O_2_/mg protein. The lipid peroxidation (LPO) was determined by measuring the MDA (a product of lipid peroxidation) content, which was measured based on the thiobarbituric acid (TBA) method [33]. The MDA content was expressed as nmol MDA/mg protein. Catalase (CAT) activity was estimated by recording the decrease in the absorbance of H_2_O_2_ at 240 nm using a UV–vis spectrophotometer (UV2150, Unico (Shanghai) Instruments Co., LTD., Shanghai, China). One unit of CAT activity (U) was defined as the quantity of enzyme required to decompose 1 μmol of H_2_O_2_ per mg of soluble protein per minute [34]. The enzyme activities were calculated per mg of protein. The activity of superoxide dismutase (SOD) was measured based on its ability to inhibit the photochemical reduction of nitroblue tetrazolium (NBT) [35]. The reaction mixture consisted of 0.3 mL of phosphate buffer (0.05 M, pH = 7.8), 0.3 mL of methionine (130 mM), 0.3 mL of NBT (0.75 mM), 0.3 mL of edetate disodium (0.1 mM),0.3 mL of riboflavin (0.02 mM), and 1.5 mL of enzymatic extract. The reaction mixture was incubated under visible light conditions for 20 min at room temperature. The absorbance was measured at 560 nm using a UV–vis spectrophotometer. One unit of SOD activity (U) was defined as the quantity of enzyme required to inhibit the photochemical reduction of NBT by 50%.

### 2.7. Statistical Analysis 

All statistical analyses were carried out using SPSS 17.0 (SPSS Inc., Chicago, IL, USA). The data are expressed as the mean values with standard deviation values (mean ± SD) of triplicate treatments. Data were analyzed using one-way analysis of variance (ANOVA) and differences were considered statistically significant at *p* < 0.05.

## 3. Results and Discussion

### 3.1. Growth Characteristics and Lipid Accumulation of Microalgae under Different Nanoparticle Concentrations

The biomass of *Chlorella* sp. UJ-3 increased with the addition of Fe_3_O_4_ NPs in a certain concentration range, but further increases in the concentration of Fe_3_O_4_ NPs resulted in a decrease in biomass (Figure 2). Pádrová et al. also found similar trends in experiments on various microalgae with zero-valent iron nanoparticles [4]. High concentrations of nanoparticles have been shown to inhibit algal biomass growth, probably because the addition of nanoparticles creates a shadowing effect that blocks photosynthetic electron transport and affects photosynthesis, ultimately affecting the growrth of algal cells [36,37]. The total lipid content of *Chlorella* sp. UJ-3 did not change significantly at low concentrations of Fe_3_O_4_ NPs, while at high concentrations of Fe_3_O_4_ NPs the total lipid content increased significantly (Figure 2). Although the biomass of algal cultures decreased at higher concentrations of Fe_3_O_4_ NPs, the total lipid production increased due to the increase in total lipid content. However, the total lipid production was lower at 200 mg/L of Fe_3_O_4_ NPs because of the sharp decrease in algal biomass. The composition and content of fatty acids are shown in Appendix A. It was found that *Chlorella* sp. UJ-3 has a variety of fatty acids, in which the saturated (C16:0 and C18:0), monounsaturated (C16:1 and C18:1), and polyunsaturated (C18:2, C18:3, C20:5) fatty acids are the major fatty acids in *Chlorella* sp. UJ-3. The results showed that the composition and content of fatty acids were almost unchanged when algal cells were exposed to low concentrations of Fe_3_O_4_ NPs. However, the polyunsaturated fatty acids (PUFA) content of *Chlorella* sp. UJ-3 increased when higher concentrations of Fe_3_O_4_ NPs were applied (Figure 3).

### 3.2. Effects of Low Concentration NPs on the Growth of Chlorella sp. UJ-3

In the low-concentration range (0–40 mg/L) tested, the biomass of algal cells first increased and then decreased with the increase in the concentration of Fe_3_O_4_ NPs, and the algal culture had the highest biomass at 20 mg/L of Fe_3_O_4_ NPs (Figure 4). The biomass increased by 52.9% and the total lipid production increased by 65.3% compared to the control without Fe_3_O_4_ NPs. The promotion of algal growth might be related to the dissociation of trace ions of nanoparticles [5]. The increased growth of microalgae supports the hypothesis that nanoparticles represent an appropriate source of trace elements [38]. Iron is essential for plant growth and metabolism, as it is associated with photosynthetic electron transport chains [26]. Trace concentrations of zero-valent iron nanoparticles (nZVI) caused the overproduction of biomass and lipids during the cultivation of cyanobacterial and microalgae [4]. The authors believed that nZVI could provide a suitable source of iron and induce biomass growth and metabolic changes, resulting in increased lipid production and changes in fatty acid composition [4]. However, the change of dissolved iron in the culture medium showed that no obvious increased dissolved iron was observed in the culture medium after Fe_3_O_4_ NPs addition. There was enough iron for cell growth since more than 40% iron remained in the culture medium (Appendix A). Iron supply was not the reason for biomass enhancement in this study. The iron recovery rate of Fe_3_O_4_ NPs also proved that no digestion of Fe_3_O_4_ NPs occurred during the culture (Appendix A). In addition, Jeon et al. demonstrated that the growth rate and fatty acid methyl ester (FAME) productivity of microalgae could be enhanced by improving the gas–liquid mass transfer rate through the addition of nanoparticles [11]. In the delayed biomass growth period (0–9 days) at 20 mg/L NPs, the effect of nanoparticles on biomass growth was not obvious, but after entering the logarithmic growth period (after nine days) the biomass concentration of the treated cells gradually became different to that of untreated cells and continued to increase, eventually reaching about 1.2 g/L (dry cell weight), which was about 54.8% higher than that of the control (Figure 5). This indicated that low concentrations of Fe_3_O_4_ NPs can improve algal biomass growth. The total lipid content of *Chlorella* sp. UJ-3 exposed to 20 mg/L Fe_3_O_4_ NPs did not change much throughout the whole growth period. The lipid production showed a gradual increase due to the increased biomass. The total lipid content and fatty acid composition did not change significantly under the concentration of Fe_3_O_4_ NPs tested (Figure 4 and Figure 6, Appendix A).

### 3.3. Effects of High Concentration NPs on Lipid Accumulation in Chlorella sp. UJ-3

The biomass and total lipid content of *Chlorella* sp. UJ-3 at high concentrations of Fe_3_O_4_ NPs are shown in Figure 7. As the concentration of Fe_3_O_4_ NPs increased, the biomass of algae decreased gradually. The addition of the Fe_3_O_4_ NPs increased the lipid content of *Chlorella* sp. UJ-3, which reached a maximum value at 100 mg/L of Fe_3_O_4_ NPs. Compared with the control algal cultures, the total lipid content increased by 42.9% and the total lipid production increased by 71.7%. However, cultures with more than 120 mg/L of Fe_3_O_4_ NPs ended up with lower lipid production due to their loss of biomass (Figure 7). The addition of nanoparticles could potentially stimulate a significant increase in lipid content in microalgae, as stress conditions can induce the accumulation of lipids in oily microalgae as a source of energy storage [17].

The content and composition of fatty acids of *Chlorella* sp. UJ-3 are summarized in Appendix A. At the Fe_3_O_4_ NP concentration of 100 mg/L, the total fatty acid content was the highest, being 235.33 ± 3.49 mg/g of dry cell weight, representing an increase of 50.6% compared with normal cultured cells. As the concentration of Fe_3_O_4_ NPs increased, the content of palmitic acid (16:0) increased slightly as the main saturated fatty acid (SFU) in *Chlorella* sp. UJ-3. The main monounsaturated fatty acid (MUFA) is oleic acid (18:1), whose content showed a downward trend (Appendix A). In contrast, the content of polyunsaturated fatty acids (PUFA) in *Chlorella* sp. UJ-3 increased gradually (Figure 8). This increase may be related to the self-protection mechanism of algal cells. When exposed to 100 mg/L of Fe_3_O_4_ NPs, the percentage of UFA in microalgae increased, the percentage of MUFA decreased, and the percentage of SFA slightly increased compared with that of untreated microalgal cells (Figure 8). This phenomenon may be due to the oxidative stress induced by Fe_3_O_4_ NPs. Under conditions of oxidative stress, a variety of free radicals produced by cells can affect various physiological functions. PUFAs have the potential for free radical scavenging, which helps to protect cells from further increases in the reactive oxygen species (ROS) level [39]. This indicated that a high accumulation of PUFAs may play an important role in the protective mechanism.

The initial total lipid content of the algal cells was approximately 19% of dry cell weight and the content of treated cells (100 mg/L) continued to increase significantly during the culture period, eventually reaching 42.1% of dry cell weight, while the control was only 29.2% of dry cell weight (Figure 9). The increase in the biomass of the algal cultures treated with Fe_3_O_4_ NPs at the later stage may have contributed by the accumulation of lipids. Since the total lipid content of the algal cells was significantly increased, the lipid production of the treated algae was significantly higher than that of the control. Similar results were found in a study of other nanoparticles [40]. Kang et al. demonstrated that a higher amount of lipids was accumulated after the exposure of *C. vulgaris* UTEX 265 to 0.1 g/L of TiO_2_ nanoparticles [40]. 

### 3.4. NPs Induced Oxidative Stress and Cellular Antioxidant Defenses

In many metabolic reactions, including photosynthesis and respiration, plant cells produce reactive oxygen species (ROS), such as superoxide radicals (O_2_^−^), singlet oxygen (^1^O_2_), hydroxyl radicals (HO•), and hydrogen peroxide (H_2_O_2_) [41]. ROS are produced as by-products of photosynthetic electron transport in photosynthesis. ROS can also be produced in mitochondria, peroxisomes, and plasma membranes by NADPH oxidases [42]. In addition, ROS are also generated during cell division, differentiation, growth, and apoptosis [43]. To control the ROS in cells, cells have evolved into an ROS scavenging system of enzymatic and nonenzymatic antioxidants, such as catalase (CAT), superoxide dismutase (SOD), peroxidase (POD), ascorbate peroxidase (APX), ascorbic acid (AsA), glutathione, tocopherol, carotenoid, etc. [43,44]. Plants and algae can activate multiple defense systems simultaneously to effectively remove different ROS.

Adverse conditions, such as glare, drought, salinity, and nutritional restrictions, have been shown to enhance the production of ROS in plant cells [44]. Nanomaterials can induce the production of reactive oxygen species (ROS), triggering the mechanism of oxidative stress in cells [45]. To assess the level of oxidative stress induced by Fe_3_O_4_ NPs, the intracellular H_2_O_2_ and malondialdehyde (MDA) content were determined during the growth of *Chlorella* sp. UJ-3. As a kind of superoxide free radical, H_2_O_2_ will accumulate in the body and cause damage to the organism when subjected to external stress. By regulating the activity of antioxidase, the organism will eliminate excess H_2_O_2_ to a stable and tolerable level, thereby reducing the threat and damage to organisms. If ROS in algal cells cannot be cleared quickly, it will cause lipid peroxidation, which will lead to the membrane dysfunction. As a product of lipid peroxidation, MDA can be used to determine the lipid peroxidation and oxidative damage of cells. The content of H_2_O_2_ and MDA in normal cultured *Chlorella* sp. UJ-3 initially increased gradually with the incubation time, reached a maximum during logarithmic growth, and then decreased gradually (Figure 10a,b). A similar trend in MDA expression of *Chlorella vulgaris* was also observed by Zhao et al. (2017) [46]. During biomass growth, the metabolic reactions of algal cells themselves produce ROS, the level of which is highest in the logarithmic growth phase. Then, as the antioxidant defense systems of cells were activated, the cells produced antioxidant enzymes and antioxidants to regulate the ROS levels. SOD and CAT, two key antioxidant enzymes, play important roles in scavenging excessive ROS. SOD is a key enzyme that removes free radicals in living organisms. It can catalyze the disproportionation reaction of excessive ROS in biological cells to generate hydrogen peroxide, which can be converted into non-toxic and harmless H_2_O and O_2_ by CAT so as to scavenge ROS and protect cells from their stress [47]. The change in enzyme activities in algal cells was similar to that of ROS (Figure 10c,d). Increases in CAT and SOD activities have been suggested to be an adaptive trait that possibly helps to overcome tissue damage by reducing ROS levels [48].

Exposure to nanoparticles resulted in an increase in ROS levels throughout the whole culture period of the algal cells. On the 12th day, the H_2_O_2_ and MDA contents of algal cells exposed to low concentrations (20 mg/L) of Fe_3_O_4_ NPs were increased by 42.9% and 51.2%, respectively, compared to untreated algal cells (Figure 10a,b), indicating that oxidative stress was generated in the cells. At this time, algal cells attempted to attenuate the effects of ROS by increasing their antioxidant enzyme activity. Compared with untreated algal cells, the activities of CAT and SOD increased by 56.7% and 61%, respectively (Figure 10c,d). The increase in antioxidant enzyme activity can substantially offset the increase in ROS induced by the nanoparticles. Cells do not need to be cleared of ROS by other means, so low concentrations of Fe_3_O_4_ NPs have little effect on the lipid content and fatty acid composition of algal cells. At high concentrations (100 mg/L) of Fe_3_O_4_ NPs, compared with the control, the contents of H_2_O_2_ and MDA and the activities of CAT and SOD were increased by 131.3%, 136.4%, 88.3%, and 95.3%, respectively (Figure 10). The activities of CAT and SOD was significantly increased, which may have been caused by the excessive production of ROS in algal cells under stress as the concentration of Fe_3_O_4_ NPs increased, thus increasing the content of antioxidant enzymes in algal cells [49]. These increased amounts of antioxidant enzyme were insufficient to remove the ROS induced by the nanoparticles, and the algal cells exposed to 100 mg/L of Fe_3_O_4_ NPs at this time may accumulate PUFAs with free radical scavenging ability through the nonenzymatic antioxidant system to scavenge excess ROS.

### 3.5. Effects of Low-High Concentration NPs Treatment on Growth and Lipid Accumulation of Chlorella sp. UJ-3

For the *Chlorella* sp. UJ-3 exposed to 20 mg/L of Fe_3_O_4_ NPs, the specific growth rate reached its maximum value on the 12th day (Figure 11). The specific growth rates for algal cell without NPs treatment or with the exposure to 100 mg/L of Fe_3_O_4_ NPs also changed with the time course and reached a maximal specific growth rate at a certain time. In addition, the maximal specific growth rate at 20 mg/L of NPs was higher than those without treatment or at 100 mg/L of Fe_3_O_4_ NPs (Appendix A). In order to obtain high-yield and high-lipid algae, the algal cells were first exposed to a low concentration of Fe_3_O_4_ NPs (20 mg/L) for 12 days, then the concentration of Fe_3_O_4_ NPs was increased to 100 mg/L when the specific growth rate reached the maximum. Additionally, the algal cells turned the metabolism towards the accumulation of lipids, which allowed large amounts of lipids to accumulate, and the total lipid production was approximately doubled compared to the untreated algal culture (Figure 12). The effect of different transition time on the combination strategy also indicated that low-high transition of Fe_3_O_4_ NPs concentration on day 12 provided the best biomass and lipid production (Appendix A). This strategy not only ensured the increase in microalgae biomass, but also promoted the accumulation of lipids. The composition and content of fatty acids are shown in Table 1. The biomass, total lipid content, and total fatty acid production increased by 50%, 42.9%, and 116.7%, respectively, compared to the control algal cells (Table 2). Furthermore, the maximum specific growth rate (µ_max_) and maximum lipid productivity (P_max_) were also calculated and were found to increase by 23.1% and 133.5%, respectively, compared with the control (Table 2). The results showed that this strategy is effective and useful for high lipid production by *Chlorella* sp. UJ-3.

## 4. Conclusions

Fe_3_O_4_ nanoparticles were found to increase the biomass and lipid production of *Chlorella* sp. UJ-3 under test conditions. When the algal cells were exposed to 20 mg/L and 100 mg/L of Fe_3_O_4_ NPs, a better biomass production and a higher lipid content were obtained, respectively. Based on these results, a new strategy was established to increase the lipid production of *Chlorella* sp. UJ-3 using Fe_3_O_4_ NPs, where the algal cells were initially exposed to low concentrations of Fe_3_O_4_ NPs and then treated with increased concentrations of Fe_3_O_4_ NPs after the maximal specific growth rate, was achieved. This strategy could potentially increase the total lipid production of algal cells by 108.7% compared to untreated algal cells. In addition, Fe_3_O_4_ NPs can affect the activity of antioxidant enzymes and the composition of fatty acids in algal cells. This study provided useful information on the biological effects of nanoparticles. At the same time, these results indicated that nanoparticles can be considered as an effective enhancer of the lipid production of microalgae using a suitable strategy. Environmental safety assessments and economic analyses of the application of this strategy in microalgae culture remain to be completed in further studies.

## Figures and Tables

**Figure 1 nanomaterials-11-02802-f001:**
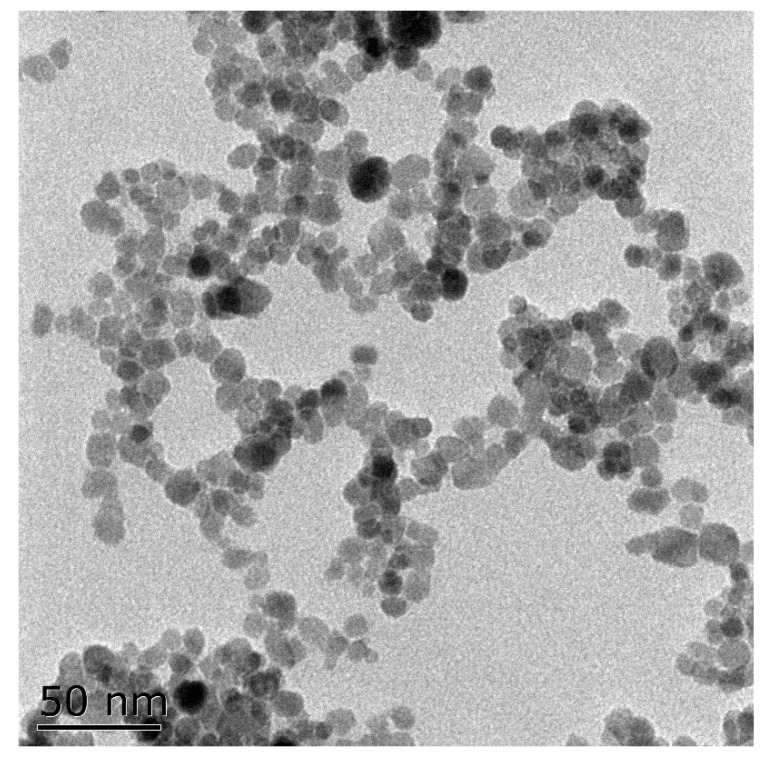
Transmission electron microscopic image of Fe_3_O_4_ NPs.

**Figure 2 nanomaterials-11-02802-f002:**
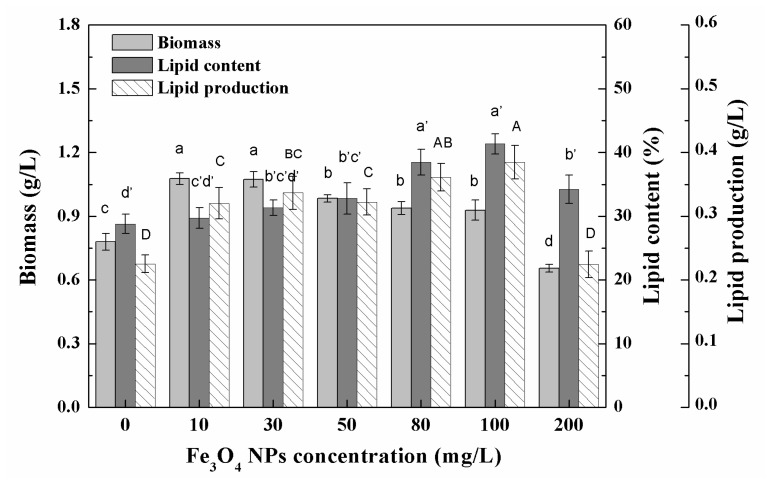
Biomass, total lipid contents, and lipid production of *Chlorella* sp. UJ-3 exposed to various concentrations of Fe_3_O_4_ nanoparticles (Culture time: 21 days; Different letters on the top of column indicate significant differences between means: a, b, c, d; a’, b’, c’, d’; A, B, C, D; *p* < 0.05).

**Figure 3 nanomaterials-11-02802-f003:**
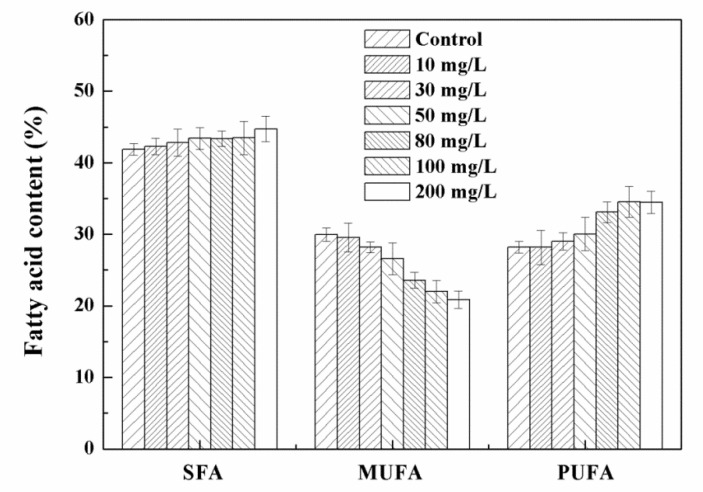
Fatty acid contents and components of *Chlorella* sp. UJ-3 exposed to different concentrations of Fe_3_O_4_ NPs (Culture time: 21 days).

**Figure 4 nanomaterials-11-02802-f004:**
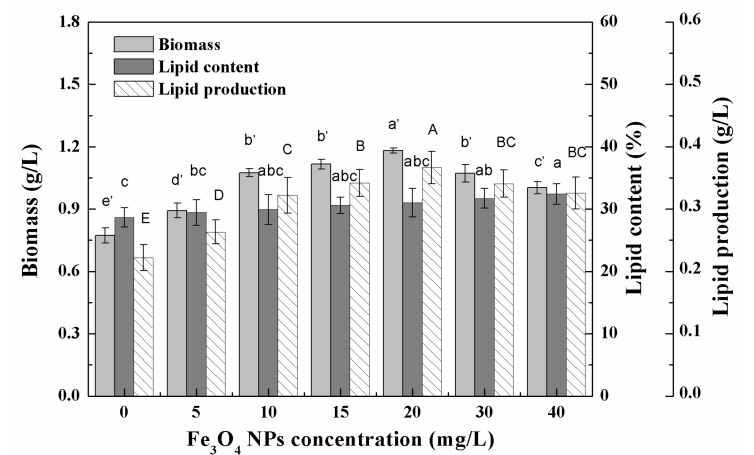
Biomass, total lipid contents, and lipid production of *Chlorella* sp. UJ-3 exposed to various concentrations of Fe_3_O_4_ NPs (Culture time: 21 days; Different letters on the top of column indicate significant differences between means: a, b, c, d; a’, b’, c’, d’; A, B, C, D; *p* < 0.05).

**Figure 5 nanomaterials-11-02802-f005:**
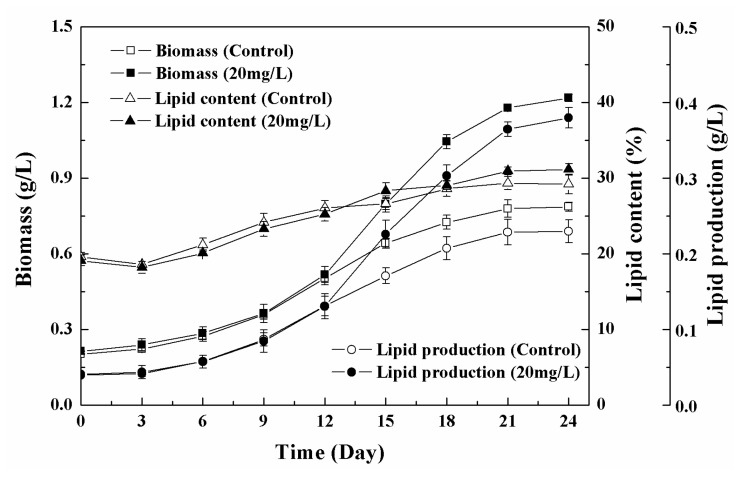
Change in biomass, total lipid contents, and lipid production of *Chlorella* sp. UJ-3 exposed to 20 mg/L Fe_3_O_4_ NPs.

**Figure 6 nanomaterials-11-02802-f006:**
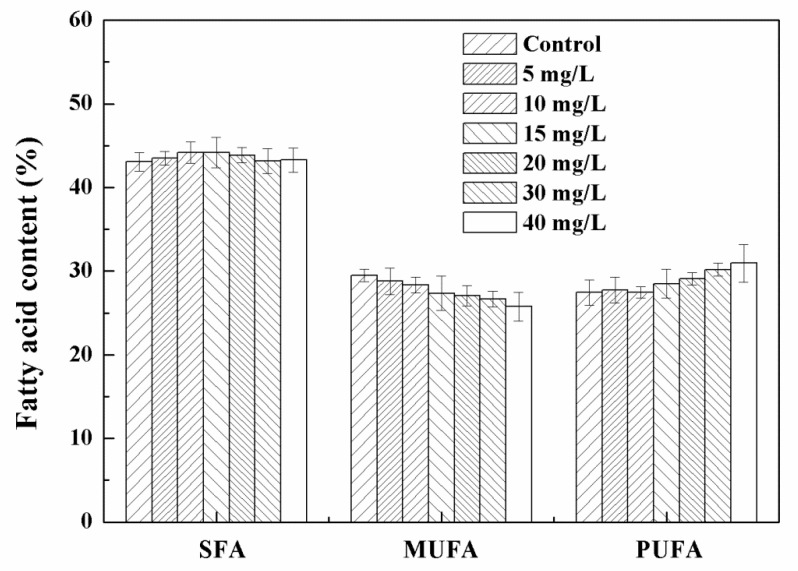
Fatty acid contents and components of *Chlorella* sp. UJ-3 exposed to different concentrations of Fe_3_O_4_ NPs (Culture time: 21 days).

**Figure 7 nanomaterials-11-02802-f007:**
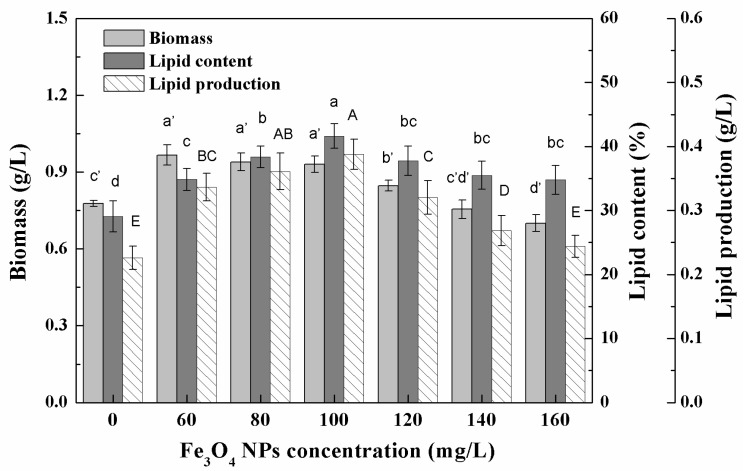
Biomass, total lipid content, and lipid production of *Chlorella* sp. UJ-3 exposed to various concentrations of Fe_3_O_4_ NPs (Culture time: 21 days; Different letters on the top of column indicate significant differences between means: a, b, c, d; a’, b’, c’, d’; A, B, C, D; *p* < 0.05).

**Figure 8 nanomaterials-11-02802-f008:**
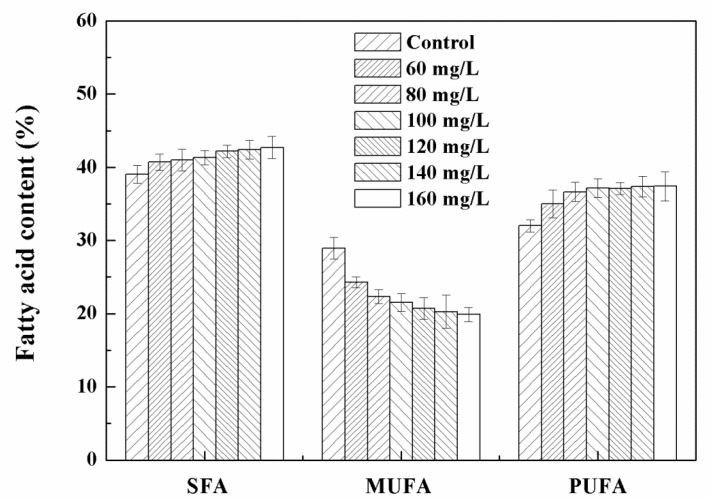
Fatty acid contents and components of *Chlorella* sp. UJ-3 exposed to different concentrations of Fe_3_O_4_ NPs (Culture time: 21 days).

**Figure 9 nanomaterials-11-02802-f009:**
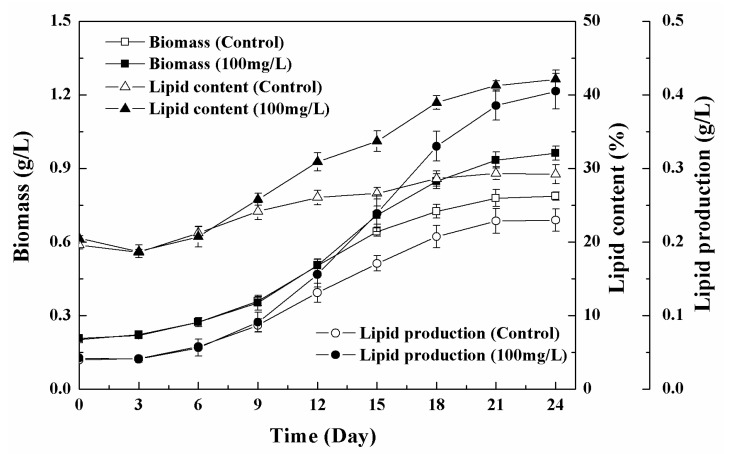
Change in biomass, total lipid content, and lipid production of *Chlorella* sp. UJ-3 exposed to 100 mg/L of Fe_3_O_4_ NPs.

**Figure 10 nanomaterials-11-02802-f010:**
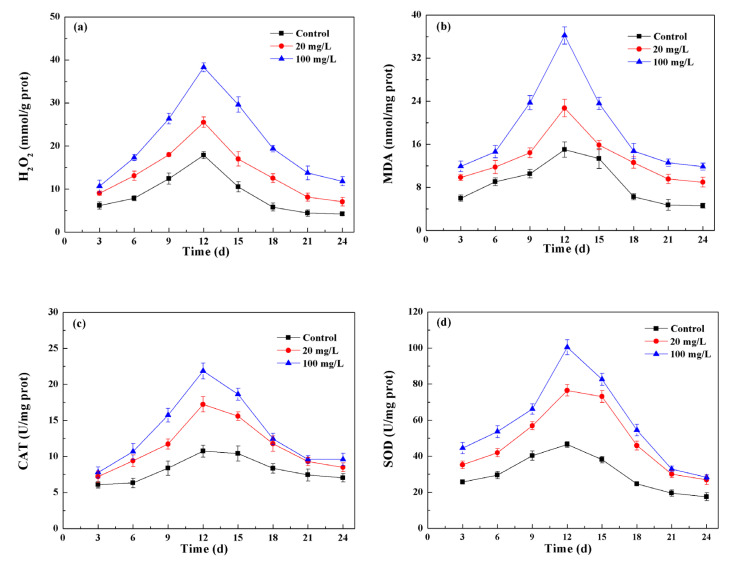
H_2_O_2_ (**a**) and MDA (**b**) contents and CAT (**c**) and SOD (**d**) activity of *Chlorella* sp. UJ-3 exposed to 20 mg/L and 100 mg/L of Fe_3_O_4_ NPs during 24 days of cultivation.

**Figure 11 nanomaterials-11-02802-f011:**
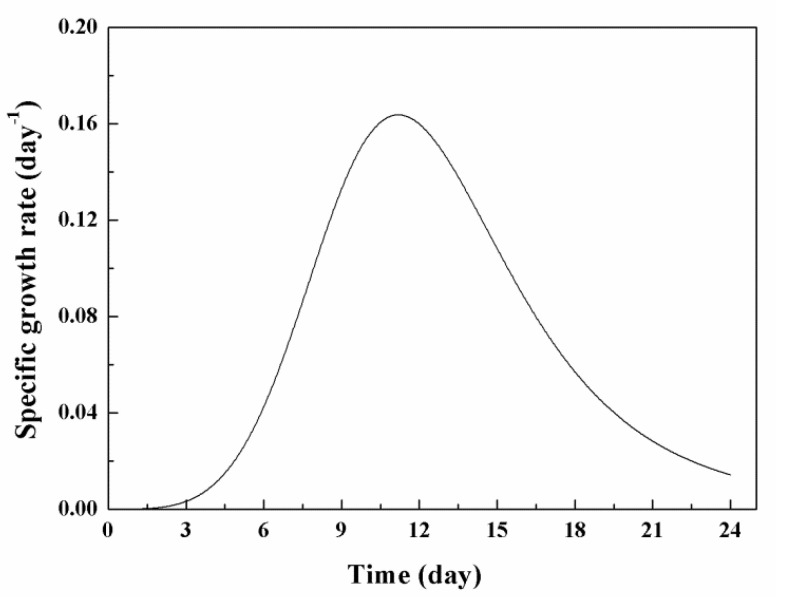
The specific growth rate of *Chlorella* sp. UJ-3 cells at 20 mg/L of NPs.

**Figure 12 nanomaterials-11-02802-f012:**
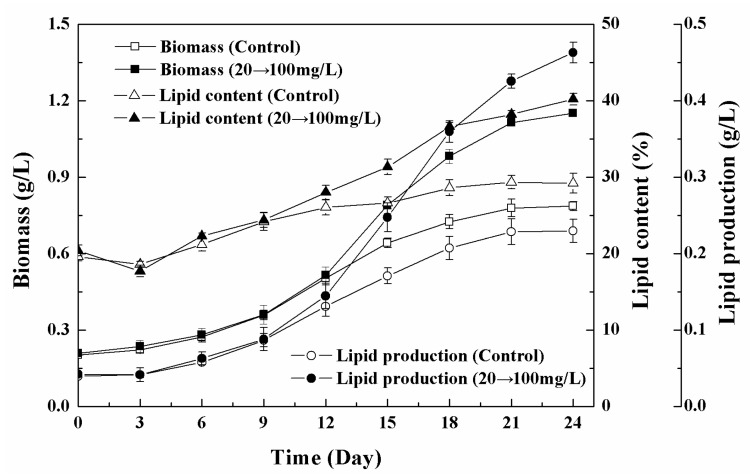
The effect of the concentration change of NPs from low (20 mg/L) to high (100 mg/L) on the growth, lipid content, and lipid production of *Chlorella* sp. UJ-3 (NP concentration change was conducted on day 12).

**Table 1 nanomaterials-11-02802-t001:** Fatty acid contents and compositions of *Chlorella* sp. UJ-3 under control conditions and NP stress.

Condition	Control		NPs Stress	
Fatty Acids	Content (mg/gDW)	Composition (%Total FA)	Content (mg/gDW)	Composition (%Total FA)
C12:0	2.07 ± 0.08	1.35 ± 0.22	3.42 ± 0.10	1.55 ± 0.25
C14:0	1.32 ± 0.14	0.86 ± 0.04	0.98 ± 0.03	0.45 ± 0.02
C16:0	54.12 ± 1.52	35.20 ± 0.38	72.68 ± 1.89	32.98 ± 0.55
C16:1	2.96 ± 0.23	1.92 ± 0.08	5.80 ± 0.43	2.63 ± 0.09
C18:0	6.29 ± 0.79	4.09 ± 0.14	18.78 ± 1.65	8.52 ± 0.92
C18:1	41.13 ± 0.67	26.75 ± 0.24	40.87 ± 1.74	18.55 ± 0.82
C18:2	6.59 ± 0.54	4.29 ± 0.23	8.47 ± 1.20	3.85 ± 0.78
C18:3n6	3.15 ± 0.42	2.05 ± 0.15	9.61 ± 0.70	4.36 ± 0.24
C18:3n3	25.24 ± 1.34	16.41 ± 0.85	36.69 ± 1.55	16.65 ± 0.49
C20:0	0.22 ± 0.05	0.14 ± 0.04	0.38 ± 0.03	0.17 ± 0.01
C20:1	0.56 ± 0.10	0.36 ± 0.03	2.25 ± 0.15	1.02 ± 0.07
C20:5	9.24 ± 0.43	6.01 ± 0.22	18.81 ± 0.86	8.83 ± 0.40
C22:0	0.88 ± 0.05	0.57 ± 0.03	1.64 ± 0.07	0.75 ± 0.03
Total	153.77 ± 3.45	100.00	220.39 ± 4.38	100.00

**Table 2 nanomaterials-11-02802-t002:** Biomass, lipid content, lipid production, maximum specific growth rate (µ_max_), and maximum lipid productivity (P_max_) of *Chlorella* sp. UJ-3 under control and NP stress.

Condition	Control	NPs Stress
Biomass (g/L)	0.78 ± 0.01	1.17 ± 0.02
Total Lipid content (% DW)	28.85 ± 2.05	41.22 ± 1.62
Lipid production (g/L)	0.23 ± 0.02	0.48 ± 0.03
µ_max_ (d^−1^)	0.13 ± 0.01	0.16 ± 0.01
P_max_ (mg L^−1^ d^−1^)	15.65 ± 0.12	36.55 ± 0.18
Fatty acids production (g/L)	0.12 ± 0.02	0.26 ± 0.02
SFA composition (%)	42.21 ± 0.44	44.42 ± 2.57
MUFA composition (%)	29.04 ± 2.26	22.20 ± 0.71
PUFA composition (%)	28.76 ± 0.79	33.39 ± 0.62

## Data Availability

The data are contained within the article and the Appendix A.

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
