# Peer review of "Development of a Strategy for Enhancing the Biomass Growth and Lipid Accumulation of *Chlorella* sp. UJ-3 Using Magnetic Fe_3_O_4_ Nanoparticles"

_nanomaterials, 2021, doi:10.3390/nano11112802_

Round 1

Reviewer 1 Report

This study investigated the mechanism underlining magnetic Fe3O4 nanoparticles. The rational behind the experiment was clear and straight forward. The manuscript is almost well written. 

There are some minor grammar issues that should be fixed in order to aid the accessibility of the results to the reader.

Author Response

We are grateful for the helpful suggestions from the reviewers. With respect to specific requests for clarification, the following revisions have been made.

Point 1: This study investigated the mechanism underlining magnetic Fe3O4 nanoparticles. The rational behind the experiment was clear and straight forward. The manuscript is almost well written.

Response 1: Thanks for your kind comment.

Point 2: There are some minor grammar issues that should be fixed in order to aid the accessibility of the results to the reader.

Response 2: Thanks for your kind suggestion. The English of this manuscript has been edited by the language service provided by MDPI (English-Editing-Certificate-35279).

Reviewer 2 Report

The authors tested the effect of Fe3O4 nanoparticles on Chlorella biomass production, lipid accumulation and stress induction under batch photoautotrophic cultivations.

The manuscript is written very well. The aims are clear and the results found are quite clear as well.

The main issue I found is that the mechanism by which Fe3O4 nanoparticles affected biomass production was not studied. Based on the results, on the methodology applied and also based on the discussion provided by the authors, is seems likely that the main mechanism by which Fe3O4 enhanced biomass production was just by providing the micronutrient Fe to the cells.
I think this is the most likely explanation. It should be in fact considered that BG11 is not the ideal growth medium for Chlorella, and iron become usually quickly deficient in BG11 for Chlorella growth.
I don’t think this can justify any practical advantage for using Fe3O4 as biomass production enhancer, because Fe deficiency can be just solved by supplying more Fe salts directly to the culture medium, without the needing to waste energy and time for producing Fe3O4 nanoparticles.
Yet, also the positive effect on lipid accumulation, seems only a result of cellular stress induction, that likely stops cell duplication. Currently this effect can be easily achieved in real cultivation plants just by applying N-deficiency, which is more easy and economic applicable than  Fe3O4 nanoparticle addition.

So, in general, my overview is that although the authors find some effects that are interesting, these effects will not give any practical advantage in real cultivation plants, because there are better alternative approached to achieve the same effects.

In the following there are some specific comments.

  • The authors should discuss the results by considering better Fe deficiency. For instance, the authors can look in literature what is the optimal Fe supply for Chlorella, and check if the BG11 was deficient for Fe for sustaining the biomass production achieved. This is an easy check that can verify the statement that Fe3O4 enhanchement is likely  due to Fe supply.
  • In the title, the word “efficient” should be removed, because there is not any explanation in the manuscript about the efficiency of the process developed, especially when compared to conventional methods.
  • Line 116-120. Please explain this section more clearly in order to make the experiments reproducible.
  • The authors should explain clearly in methods how they calculated Pmax and µmax (indicating also what experimental points were used).
  • Figure 11. In the manuscript there is not written how the authors obtained such a curve for the growth rate. This should be clearly explained.
  • “cell growth” should be replaced throughout manuscript by “biomass growth”, because the authors did not count cells.
  • Chlorella” should always be written with the first letter as capital letter, check it in the text to fix mistakes.
  • Table 5. what does <0.01 mean?
  • Line 398-400. This sentence should be removed because your work did not provide any finding about the improvement by magnetic harvesting.

Author Response

We are grateful for the helpful suggestions from the reviewers. With respect to specific requests for clarification, the following revisions have been made.

Point 1: The authors tested the effect of Fe3O4 nanoparticles on Chlorella biomass production, lipid accumulation and stress induction under batch photoautotrophic cultivations.The manuscript is written very well. The aims are clear and the results found are quite clear as well.

Response 1: Thanks for your kind comment.

Point 2: The main issue I found is that the mechanism by which Fe3O4 nanoparticles affected biomass production was not studied. Based on the results, on the methodology applied and also based on the discussion provided by the authors, is seems likely that the main mechanism by which Fe3O4 enhanced biomass production was just by providing the micronutrient Fe to the cells.I think this is the most likely explanation. It should be in fact considered that BG11 is not the ideal growth medium for Chlorella, and iron become usually quickly deficient in BG11 for Chlorella growth.I don’t think this can justify any practical advantage for using Fe3O4 as biomass production enhancer, because Fe deficiency can be just solved by supplying more Fe salts directly to the culture medium, without the needing to waste energy and time for producing Fe3O4 nanoparticles.Yet, also the positive effect on lipid accumulation, seems only a result of cellular stress induction, that likely stops cell duplication. Currently this effect can be easily achieved in real cultivation plants just by applying N-deficiency, which is more easy and economic applicable than Fe3O4 nanoparticle addition.So, in general, my overview is that although the authors find some effects that are interesting, these effects will not give any practical advantage in real cultivation plants, because there are better alternative approached to achieve the same effects.

Response 2: Thanks for your kind suggestion. For the mechanism of enhanced biomass production by Fe3O4 nanoparticles, more data about the dissolved Fe content in medium was tested along with the culture time and the data was added in supplementary materials. The results indicated the Fe supply was not the reason for the improvement in biomass production and the related results and discussion was added in the content of manuscript. Therefore, more study about the positive effect of Fe3O4 nanoparticles on algal biomass production will be studied in future. In fact, the Fe3O4 nanoparticles also showed the similar effect on the biomass production of Nannochloropsis maritime and Botryococcus braunii. Therefore, we hope to find the detailed mechanism about positive effect of Fe3O4 nanoparticles after we investigated different algal cells in further study. BG11 medium is an effective medium for the cell growth of microalgae and was widely used in Chlorella cultivation.

Fe3O4 nanoparticles were cheap commercial product now. And also, we can prepare it easily. Therefore, the cost of the use of Fe3O4 nanoparticles was low. Another reason for the selection of Fe3O4 nanoparticles as the inducer for biomass and lipid production of microalgae is that we have developed efficient methods and apparatus for magnetic separation of microalgae. Therefore, we hope establish a strategy to improve the biomass and lipid production using magnetic nanoparticles and then followed by the magnetic separation. We think it will be a useful method for the lipid production by microalgae with high lipid production and easier harvesting operation. So, this study in this manuscript was very important and has a good potential in future application. Our published paper about magnetic separation was listed as follows:

  • Hu YR, Guo C, Xu L, Wang F, Wang SK, Hu ZM, Liu CZ. A magnetic separator for efficient microalgae harvesting. Bioresource Technology, 2014, 158: 388–391
  • Wang SK, Wang F, Hu YR, Stiles AR, Guo C, Liu CZ. Magnetic flocculant for high efficiency harvesting of microalgal cells. ACS Applied Materials & Interfaces, 2014, 6: 109-115
  • Hu YR, Guo C, Wang F, Wang SK, Pan F, Liu CZ. Improvement of microalgae harvesting by magnetic nanocomposites coated with polyethylenimine. Chemical Engineering Journal, 2014, 242: 341–347
  • Hu YR, Wang F, Wang SK, Liu CZ, Guo C. Efficient harvesting of marine microalgae Nannochloropsis maritima using magnetic nanoparticles. Bioresource Technology, 2013, 138: 387-390.
  • Xu L, Guo C, Wang F, Zheng S, Liu CZ. A simple and rapid harvesting method for microalgae by in situ magnetic separation. Bioresource Technology, 2011, 102: 10047-10051.

Point 3: The authors should discuss the results by considering better Fe deficiency. For instance, the authors can look in literature what is the optimal Fe supply for Chlorella, and check if the BG11 was deficient for Fe for sustaining the biomass production achieved. This is an easy check that can verify the statement that Fe3O4 enhanchement is likely  due to Fe supply.

Response 3: Thanks for your kind suggestion. According to your suggestion, we added more data about the dissolved Fe content in medium along with the culture time and the data was provided in supplementary materials. The results indicated the Fe supply was not the reason for the improvement in biomass production and the related results and discussion were added in the Section 3.2.

Point 4: In the title, the word “efficient” should be removed, because there is not any explanation in the manuscript about the efficiency of the process developed, especially when compared to conventional methods.

Response 4: Thanks for your kind suggestion. It has been deleted.

Point 5: Line 116-120. Please explain this section more clearly in order to make the experiments reproducible.

Response 5: Thanks for your kind suggestion. It has been revised in the Section 2.3.

Point 6: The authors should explain clearly in methods how they calculated Pmax and µmax (indicating also what experimental points were used).

Response 6: Thanks for your kind suggestion. The methods for the calculations of Pmax and µmax have been provided in the Section 2.4.

Point 7: Figure 11. In the manuscript there is not written how the authors obtained such a curve for the growth rate. This should be clearly explained.

Response 7: Thanks for your kind suggestion. It has been explained in the Section 2.4.

Point 8: “cell growth” should be replaced throughout manuscript by “biomass growth”, because the authors did not count cells.

Response 8: Thanks for your kind suggestion. It has been revised throughout manuscript.

Point 9: “Chlorella” should always be written with the first letter as capital letter, check it in the text to fix mistakes.

Response 9: Thanks for your kind suggestion. It has been revised.

Point 10: Table 5. what does <0.01 mean?

Response 10: Thanks for your kind suggestion. “<” should be “±”. It has been revised.

Point 11: Line 398-400. This sentence should be removed because your work did not provide any finding about the improvement by magnetic harvesting.

Response 11: Thanks for your kind suggestion. These sentences have been deleted since there was no data to support the conclusion in this manuscript. In fact, the magnetic separation can be carried out according to the results in our previous study [1].

[1] Xu L, Guo C, Wang F, Zheng S, Liu CZ. A simple and rapid harvesting method for microalgae by in situ magnetic separation. Bioresource Technology, 2011, 102: 10047-10051.

Reviewer 3 Report

The manuscript deals with a topic of potential interest and real practical applications, and does so by employing a sound range of experimental techniques.

A minor note may be made chemicals and equipment used are missing company and country.

Author Response

We are grateful for the helpful suggestions from the reviewers. With respect to specific requests for clarification, the following revisions have been made.

Point 1: The manuscript deals with a topic of potential interest and real practical applications, and does so by employing a sound range of experimental techniques.

Response 1: Thanks for your kind comment.

Point 2: A minor note may be made chemicals and equipment used are missing company and country.

Response 2: Thanks for your kind suggestion. It has been added in the section “Materials and Methods”. 

Reviewer 4 Report

This work aims to develop an efficient strategy for enhanced cell growth and lipid accumulation by Chlorella sp. UJ-3 using magnetic Fe3O4 nanoparticles. The authors have prepared Fe3O4 nanoparticles and given several doses of the nanoparticle (ranging from 5-200 mg/L along with control) to Chlorella sp. UJ-3 culture. It was observed that there was some increase in the algal biomass as well as lipid production. Although the work seems interesting, yet there are few issues appended below:

- Introduction mentions the previous reports on the use of nanoparticles, yet within the MS authors kept statements that were not supported by the results. Example: 1. Fe may be released and worked as micronutrients (free Fe was NOT checked, neither just Fe3O4 powders were used to confirm this. Besides, the amount of 100 mg/L is too high to be considered as MICROnutrient.

Example 2: Lipid accumulation was justified to be an indirect gain due to ROS, however, it is reported that only PUFA increases with ROS. How total lipids get increased. So, is it the MICROnutrient effect or the ROS that played the role? Nanoparticles are known to be toxic to the cells, the specific growth rate indicates this (a decline after 12 days with 20 mg/L NPs). It was not shown with 100 mg/L NP dose, why does biomass increase when a combination was used?

Example 3: Line 398-399 Magnetic NPs can be used to collect algal biomass (but it was not checked).

- Table 1,2 and 3 seems redundant.

- Figure 2, the calculation for lipid content seems erroneous, particularly for 80, 100, and 200 mg/L.

- Figure 10: Why control (untreated) shows high ROS-related activities at 12 days but it declines with time without any treatment!!.

- The internalization of NPs within algal cells, free Fe vs NP levels with time should have also been studied.

Based on the above observations, I vote against this Manuscript.

Author Response

We are grateful for the helpful suggestions from the reviewers. With respect to specific requests for clarification, the following revisions have been made.

Point 1: Introduction mentions the previous reports on the use of nanoparticles, yet within the MS authors kept statements that were not supported by the results. Example: 1. Fe may be released and worked as micronutrients (free Fe was NOT checked, neither just Fe3O4 powders were used to confirm this. Besides, the amount of 100 mg/L is too high to be considered as MICROnutrient.

Response 1: Thanks for your kind suggestion. More data about the dissolved Fe content in medium was tested along with the culture time and the data was added in supplementary materials. The related results and discussion were added in the Section 3.2. The amount of 100 mg/L Fe3O4 nanoparticles were solids and they can be recovered from the separated mixture of algal cells and Fe3O4 NPs. More results about nanoparticle recovery were provided in supplementary materials.

Point 2: Example 2: Lipid accumulation was justified to be an indirect gain due to ROS, however, it is reported that only PUFA increases with ROS. How total lipids get increased. So, is it the MICROnutrient effect or the ROS that played the role? Nanoparticles are known to be toxic to the cells, the specific growth rate indicates this (a decline after 12 days with 20 mg/L NPs). It was not shown with 100 mg/L NP dose, why does biomass increase when a combination was used?

Response 2: Thanks for your kind suggestion. It has been explained in the manuscript that algal cells can accumulate PUFAs with free radical scavenging ability through the nonenzymatic antioxidant system to scavenge excess ROS. The high dosage of 100 mg/L NP induced high ROS and the ROS induced PUFAs synthesis. And thus, the PUFAs production increased and PUFAs content in fatty acid also increased, resulting the increase of total lipid production.

The supplementary data has proved that NPs addition did not affect the MICROnutrient in medium.

It must be mentioned that the specific growth rate for algal cell without NPs treatment also change with the time course and reached a maximal specific growth rate at a certain time. This is the basic consensus. In this study, the addition of 20 mg/L NPs didn’t exhibit toxic to algal cells and it was beneficial for cell growth (Figure 5). In the combination strategy, the addition of 20 mg/L NPs was applied to achieved the higher biomass and then the high concentration of 100 mg/L NPs was used for lipid accumulation. Therefore, the final biomass was increased in the combination strategy.

Point 3: Example 3: Line 398-399 Magnetic NPs can be used to collect algal biomass (but it was not checked).

Response 3: Thanks for your kind suggestion. These sentences have been deleted since there was no data to support the conclusion in this manuscript. In fact, the conclusion was obtained according to the results in our previous study [1,2].

  • Xu L, Guo C, Wang F, Zheng S, Liu CZ. A simple and rapid harvesting method for microalgae by in situ magnetic separation. Bioresource Technology, 2011, 102: 10047-10051.
  • Hu YR, Guo C, Xu L, Wang F, Wang SK, Hu ZM, Liu CZ. A magnetic separator for efficient microalgae harvesting. Bioresource Technology, 2014, 158: 388–391

Point 4: Table 1,2 and 3 seems redundant.

Response 4: Thanks for your kind suggestion. Table 1,2 and 3 provided detailed information about the fatty acid composition and were useful for the reader. Table 1,2 and 3 were moved to the supplementary materials.

Point 5: Figure 2, the calculation for lipid content seems erroneous, particularly for 80, 100, and 200 mg/L.

Response 5: Thanks for your kind suggestion. We have checked it and the data was correct.

Point 6: Figure 10: Why control (untreated) shows high ROS-related activities at 12 days but it declines with time without any treatment.

Response 6: Thanks for your kind suggestion. In the control, ROS was produced during the cell proliferation and high cell growth rate generally caused high ROS production. In addition, algal cells possessed their own antioxidant system including enzymatic and non-enzymatic system, which will help cell to decrease the ROS level.

Point 7: The internalization of NPs within algal cells, free Fe vs NP levels with time should have also been studied.

Response 7: Thanks for your kind suggestion. More data about the dissolved Fe content in medium was tested along with the culture time and the data was added in supplementary materials. More results about Fe recovery of Fe3O4 nanoparticle were also provided in supplementary materials. These results indicated that the addition of Fe3O4 nanoparticles didn’t affect the free Fe level in medium. The internalization of NPs was not determined in this study, but the results indicated that the Fe3O4 nanoparticles was not digested by algal cells. 

Round 2

Reviewer 4 Report

The authors have addressed some of the concerns raised previously. However, there is still some issue that needs attention, as appended below:

- The sudden drop in the specific growth rate as shown in Fig. 11 is NOT the general consensus, as the plateau is too sharp, indicating a decline in the growth rate possible due to toxicity or inhibitory activity. Similar data with control and 100 mg/L NPs should also be provided.

- Author states that ‘the specific growth rate for algal cell without NPs treatment also changes with the time course and reached a maximal specific growth rate at a certain time.’ In this study, the addition of 20 mg/L NPs didn’t exhibit toxicity to algal cells and it was beneficial for cell growth (Figure 5, then why specific growth rate decreased drastically after 12 days?; see fig. 11).

- Author states that “In the combination strategy, the addition of 20 mg/L NPs was applied to achieved the higher biomass and then the high concentration of 100 mg/L NPs were used for lipid accumulation. Therefore, the final biomass increased in the combination strategy.”:  Again, this is surprising that NPs acts differently! Why is it so? How the ROS is initiating biomass growth at 20 mg but lipid at 100 mg! [Figure 7 vs Fig. 9: Biomass shown for 100 mg NPs are different in both these figures!]

- If PUFA increases during ROS response, Why SFA increased? (See fig. 8, table 1, and table 2 of the revised MS).

- Why the addition of 100 mg NP was done at 12 days and not before or after??

  • Response to Point 6 [i.e.: Figure 10: Why control (untreated) shows high ROS-related activities at 12 days but it declines with time without any treatment] was not satisfactory. Please provide reference to support this, why it took 12 days to accumulate ROS for necessary action by the cell, ROS is deadly for any cell and should be immediately stopped.

Note: Experimental details, particularly harvesting time and culture conditions should be mentioned on the figure legend itself.

Author Response

Response to Reviewer 4 Comments

We are grateful for the helpful suggestions from the reviewers. With respect to specific requests for clarification, the following revisions have been made.

Point 1: The sudden drop in the specific growth rate as shown in Fig. 11 is NOT the general consensus, as the plateau is too sharp, indicating a decline in the growth rate possible due to toxicity or inhibitory activity. Similar data with control and 100 mg/L NPs should also be provided.

Response 1: Thanks for your kind suggestion. The change of the specific growth rate of Chlorella sp. UJ-3 cells without treatment (a) and with the treatment at 100 mg/L of NPs has been provided in the supplementary materials. The similar trend was obtained. The specific growth rate for algal cell without NPs treatment also changes with the time course and reached a maximal specific growth rate at a certain time. The related results have been added in Section 3.5. The similar change of specific growth rate for Chlorella minutissima also has been reported by Amaral et al (Figure 1) [1].

(See attachment)

Figure 1 Specific growth rate of Chlorella minutissima [1]

Reference

[1] Amaral, M.S.; Loures, C.C.A.; Naves, F.L.; Baeta, B.E.L.; Silva, M.B.; Prata, A.M.R. Evaluation of cell growth performance of microalgae Chlorella minutissima using an internal light integrated photobioreactor. Journal of Environmental Chemical Engineering 2020, 8, 104200

Point 2: Author states that ‘the specific growth rate for algal cell without NPs treatment also changes with the time course and reached a maximal specific growth rate at a certain time.’ In this study, the addition of 20 mg/L NPs didn’t exhibit toxicity to algal cells and it was beneficial for cell growth (Figure 5, then why specific growth rate decreased drastically after 12 days?; see fig. 11).

Response 2: Thanks for your kind suggestion. The change of the specific growth rate of Chlorella sp. UJ-3 cells without treatment (a) and with the treatment at 100 mg/L of NPs has been provided in the supplementary materials. It proved the statement that “the specific growth rate for algal cell without NPs treatment also changes with the time course and reached a maximal specific growth rate at a certain time”. In addition, the maximal specific growth rate at 20 mg/L of NPs was higher than that obtained in the control. Therefore, it indicated that the addition of 20 mg/L NPs didn’t exhibit toxicity to algal cells and it was beneficial for cell growth. The related results have been added in Section 3.5.

Point 3: Author states that “In the combination strategy, the addition of 20 mg/L NPs was applied to achieved the higher biomass and then the high concentration of 100 mg/L NPs were used for lipid accumulation. Therefore, the final biomass increased in the combination strategy.”:  Again, this is surprising that NPs acts differently! Why is it so? How the ROS is initiating biomass growth at 20 mg but lipid at 100 mg! [Figure 7 vs Fig. 9: Biomass shown for 100 mg NPs are different in both these figures!]

Response 3: Thanks for your kind suggestion. The addition of 20 mg/L NPs was applied to achieved the higher biomass has been proved in Section 3.2 and the high concentration of 100 mg/L NPs were used for lipid accumulation has been proved in Section 3.3. Based on these results, the combination strategy was applied. More study about the positive effect of Fe3O4 nanoparticles on algal biomass production will be studied in future. In fact, the Fe3O4 nanoparticles also showed the similar effect on the biomass production of Nannochloropsis maritime and Botryococcus braunii. Therefore, we hope to find the detailed mechanism about positive effect of Fe3O4 nanoparticles after we investigated different algal cells in further study. The higher ROS can be one of the reasons for the accumulation of lipid at higher NPs concentration (Section 3.4).  Biomass in Figure 7 for 100 mg/L NPs was 0.94±0.02 g/L and Biomass in Figure 9 for 100 mg/L NPs was 0.95±0.02 g/L on Day 21. There was no significant difference for these two results.  The information about culture time has been added in Figure 7.

Point 4: If PUFA increases during ROS response, Why SFA increased? (See fig. 8, table 1, and table 2 of the revised MS).

Response 4: Thanks for your kind suggestion. The biosynthesis of PUFA and SFA shared some synthetic pathways. Recently, the transcriptome analysis was conducted under different NPs treatments in our research group. It was found that the gene expression of the shared enzyme (e.g. acyl-ACP desaturase, 3-ketoacyl ACP reductase) was upregulated under NPs treatment. This can be the reason for the increased SFA production during the ROS response.

Point 5: Why the addition of 100 mg NP was done at 12 days and not before or after?

Response 5: Thanks for your kind suggestion. The transition time for the combination strategy was also tested in different time and the results has been added in the supplementary materials. The results showed that the NPs concentration increased to 100 mg/L on day 12 provided the best biomass and lipid production.

Point 6: Response to Point 6 [i.e.: Figure 10: Why control (untreated) shows high ROS-related activities at 12 days but it declines with time without any treatment] was not satisfactory. Please provide reference to support this, why it took 12 days to accumulate ROS for necessary action by the cell, ROS is deadly for any cell and should be immediately stopped.

Response 6: Thanks for your kind suggestion. The ROS increased with the prolonged culture time and then declined after 12 days. Similar trend of MDA expression of Chlorella vulgaris was also reported by Zhao et al (2017) [2]. The highest MDA in this study was less than 40 nmol/mg after 12 days culture. In the recent reported literature, the MDA expression of Chlorella vulgaris reached more than 300 nmol/mg after 7 days culture without any treatment [3]. Therefore, the ROS level of Chlorella sp. in this study was reasonable and it was still in the range where the cell can be survived, but the cell metabolism may be changed. For example, in our recent transcriptome analysis, the results indicated the enzyme in Calvin cycle (e.g. ALDO, FBP, GAPDH) was upregulated under NPs treatment, which was beneficial for the fixation of CO2, providing more carbon source for cell growth and metabolism.

Reference

[2] Zhao, F.F.; Xiang, Q.Q.; Zhou, Y.; Xu, X.; Qiu, X.Y.; Yu, Y.; Ahmad, F. Evaluation of the toxicity of herbicide topramezone to chlorella vulgaris: oxidative stress, cell morphology and photosynthetic activity. Ecotoxicology & Environmental Safety 2017, 143, 129-135.

[3] Ma, Z.H.; Yang, F.S.; Ren, J.Y.; Fan, R.; Duan, Q.N.; Guo, J.; Guo, J.H. Growth inhibition and oxidative stress in two green algal species exposed to erythromycin. Journal of the American Water Resources Association 2021, 57, 628-637.

Point 7: Note: Experimental details, particularly harvesting time and culture conditions should be mentioned on the figure legend itself.

Response 7: Thanks for your kind suggestion. The culture condition was described in Section 2.1. The information for the culture time has been added in the caption of figures.
